# Landscape and female fertility evaluation of seven heavenly bamboo cultivars as potential non-invasive alternatives to the wildtype

**Julia Rycyna**[1]*, **Sandra Wilson**[1], **Zhanao Deng**[2], **Basil Iannone**[3], **Gary Knox**[4]

**1** Department of Environmental Horticulture, Institute of Food and Agricultural Sciences, University of Florida, Gainesville, Florida, United States of America, **2** Department of Environmental Horticulture, Institute of Food and Agricultural Sciences, Gulf Coast Research and Education Center, University of Florida, Wimauma, Florida, United States of America, **3** School of Forest, Fisheries and Geomatics Sciences, University of Florida, Gainesville, Florida, United States of America, **4** Department of Environmental Horticulture, Institute of Food and Agricultural Sciences, North Florida Research and Education Center, University of Florida, Quincy, Florida, United States of America

* juliarycyna@ufl.edu

**Data Availability Statement:** All relevant data are within the paper and its Supporting Information files.

## Abstract

In recent years, breeding initiatives have been made to reduce the fecundity of invasive plants leading to sterile cultivars. The wildtype form of heavenly bamboo (*Nandina domestica* Thunb. (Berberidaceae) and seven cultivars were evaluated for landscape performance, fruit production and seed viability at three sites in Florida located in southwest, northcentral, and north Florida. For heavenly bamboo cultivars in north Florida, 'Emerald Sea', 'Greray' (Sunray®), 'Lemon-Lime', 'Murasaki' (Flirt™), 'SEIKA' (Obsession™), and 'Twilight' performed well throughout much of the study with average visual quality ratings between 3.54 and 4.60 (scale of 1 to 5). In northcentral Florida heavenly bamboo cultivars are 'Emerald Sea', 'Greray', 'Lemon-Lime', 'Murasaki', 'SEIKA', and 'Twilight' performed well throughout much of the study with average quality ratings between 4.49 and 4.94. In southwest Florida, 'Emerald Sea', 'Greray', 'Lemon-Lime', 'Murasaki', and 'SEIKA' performed well with average quality ratings between 3.40 and 4.83. At all three sites, 'Emerald Sea' and the wildtype were similar in size, having the greatest growth indices compared to medium-sized cultivars ('Greray', 'Lemon-Lime', 'Twilight' and 'SEIKA') and dwarf-sized cultivars ('Chime' and 'Murasaki'). For three consecutive fall-winter seasons of the study, 'Chime', 'Greray', and 'Lemon-Lime' heavenly bamboo did not fruit at any of the study sites. Among the three sites, 'Murasaki' had 97.7% to 99.9% fruit reduction, 'SEIKA' had 97.7% to 100% fruit reduction, and 'Twilight' had 95.9% to 100% fruit reduction compared to the wildtype at respective sites. Seeds collected from low fruiting cultivars ('Murasaki', 'SEIKA', and 'Twilight') had 33.3% to 66.7% viability, as determined by tetrazolium tests. In comparison, 'Emerald Sea' produced as much, if not more, fruit as the wildtype, especially in northern Florida, with seed viability ranging from 6.7% to 29.0% among sites. Nuclear DNA content of cultivars were comparable to the wildtype, suggesting they are diploids. These findings identified four low to no fruiting heavenly bamboo cultivars recommended for landscape use ('Lemon-Lime', 'SEIKA', 'Murasaki', and 'Greray').

**Funding:** Financial support of this project was from the U.S. Department of Agriculture's (USDA) Agricultural Marketing Service through grant number AM180100XXXXG046 and the Florida Department of Agriculture and Consumer Services (FDACS) Specialty Crop Block Grant Program (SCBGP) contract # 025784. The funders had no role in study design, data collection and analysis, decision to publish, or preparation of the manuscript.

**Competing interests:** The authors have declared that no competing interests exist.

# Introduction

Heavenly bamboo (*Nandina domestica*) is a commonly used landscaping shrub, often chosen for its evergreen foliage, white panicles of flowers in the summer, and brilliant red berries in the colder months. Depending on the cultivar, it can be planted in borders, containers, hedges, and mass plantings and can also provide fall color in a range of burgundy-red to light pink hues. The plants do well in full sun to part shade, are tolerant of a wide range of soils, and are considered low maintenance and drought tolerant [1].

Native to central China, Japan, and India, heavenly bamboo was introduced to the United States for ornamental use in 1804. Over 150 years later it was first noted as naturalizing in North Carolina [2]. In 2000, Cherry [3] documented self-sustaining and expanding populations of heavenly bamboo altering light conditions and displacing native species in natural plant communities of northern and central Florida. Under natural conditions, seeds of heavenly bamboo berries typically develop in fall, persist through winter in leaf litter, undergo warm stratification in summer months, and then eventually germinate [4]. Thus, seeds of the species have a morphophysiological dormancy [5–7] described as a combination of morphological dormancy and physiological dormancy [8].

While the heavenly bamboo resident taxon (referred to as wildtype from this point) is still commercially available, the current nursery inventory predominately consists of cultivated selections that have been bred for improved and novel form and foliage color [4]. In fact, there are 65 named cultivars in Japan, and over 40 cultivars have been catalogued in the JC Raulston Arboretum (Raleigh, NC) [9]. As early as 2003, Wilson and her colleagues began to evaluate cultivated forms of heavenly bamboo that could potentially serve as suitable non-invasive replacements to the wildtype [4]. From two separate studies, they identified eight cultivars ('AKA', 'Firepower', 'Firestorm', 'Firehouse', 'Moon Bay', 'Gulf Stream', 'Harbour Dwarf', and 'Jaytee') that not only performed well at multiple locations in Florida, but also had greater than 98% fruit reduction at one or more locations [10,11]. An additional cultivar, 'Filamentosa', was absent of fruit, but it was less suitable for Florida's landscape conditions [11]. Other significant findings were 1) fruit production was significantly greater in northern Florida than in southern Florida, and 2) the wildtype and all cultivars evaluated were diploids. While this suggests polyploidy is not the cause of female infertility in heavenly bamboo, additional cultivars merit verification.

Since these trials, other cultivars have been released to the industry, but their invasive potential has not been evaluated in replicated research trials. Modern, reportedly berry-free cultivars (produced from whole plant mutations, sports or intentional crosses) have been marketed for their superior uniformity and extended foliage color [12]. Despite these advancements in sterile cultivar development, at the University of Florida (UF) all cultivars must be evaluated and approved as noninvasive by the UF/IFAS Infraspecific Taxon Protocol (ITP) assessment before recommending their use. This protocol consists of 12 questions to determine 1) if the cultivar can be readily distinguished from the wildtype and displays invasive traits that cause greater ecological impact than the wildtype or resident species; and 2) the fecundity of the cultivar and its chances of regression or hybridization to characteristics of the wildtype (or naturalized resident species) [13].

Cultivar evaluation of invasive potential is dynamic and increasingly important with new plant introductions. To date, five cultivars ('Firepower', 'Harbour Dwarf', 'AKA', 'Firehouse', and 'Firestorm') have been approved for use throughout Florida by the UF/IFAS ITP, and two cultivars ('Jaytee' and 'Gulf Stream') were approved for use with caution in southern Florida (but not approved in northern or central Florida) [4]. Continuing this work, the overall goal of this study was to identify additional good-performing and non-fruiting cultivars of heavenly

bamboo that could serve as viable alternatives to the wildtype. Specific objectives were to: 1) determine the effects of location and cultivar on landscape performance (visual appeal), growth and female fertility of heavenly bamboo grown in replicated trials, 2) assess the seed viability of resultant fruit, and 3) determine the nuclear DNA content to infer if the ploidy level of these cultivars differed from that of the invasive wildtype.

## Materials and methods

### Plant material and site conditions

Seven heavenly bamboo cultivars were evaluated in this study in addition to the wildtype. These cultivars included 'Chime', 'Greray', 'Lemon-Lime', 'Murasaki', 'SEIKA', 'Twilight', and 'Emerald Sea'; their characteristics are described in Table 1. Plants were obtained as finished 11.4-L plants (Greenleaf Nursery Co., El Campo, TX; Monrovia Nursery Co., Azusa, CA; and May Nursery, Havana, FL) except for 'Emerald Sea', which was received as liners and finished as 2.84-L plants prior to planting.

Experiments were conducted at three locations located in southwest FL [Gulf Coast Research and Education Center (GCREC), Balm], northcentral FL [Plant Science Research and Education Unit (PSREU), Citra], and north FL [North Florida Research and Education Center (NFREC), Quincy]. Before planting, beds were prepared by applying glyphosate herbicide (Roundup; Monsanto Co., St. Louis, MO) and slightly disked before covering with black semipermeable landscape fabric (Lumite Inc., Baldwin, GA). In northcentral FL, ground beds were treated with a multipurpose liquid fumigant (Pic-Clor 60; active ingredients 1,3- dichloropropene and chloropicrin) 3 weeks before planting. Taxon were spaced 1.2 m on center under full sun conditions in three locations. Plants were initially watered with drip-irrigation twice a day for 35–60 mins. Once established, the irrigation was reduced to three times a week. All plants were fertilized with approximately 84 g of 15N–3.9P–10K 8–9 month controlled-release fertilizer (Osmocote Plus; Scotts, Marysville, OH) in the area 15 to 30 cm from the crown. Plants were top-dressed with Osmocote every six months.

**Table 1. Botanical description of heavenly bamboo (*Nandina domestica*) cultivars and wildtype evaluated for landscape performance, growth, fruiting, and ploidy level.**

| Taxon | Common Name | Size Category[z] | Description |
|---|---|---|---|
| Heavenly bamboo 'Chime' | Chime | Dwarf | Compact mounded form with thread-like, chartreuse green finely dissected foliage that turns orange-red in winter. |
| Heavenly bamboo 'Emerald Sea' | Emerald Sea | Large | Upright habit with emerald-green foliage having a purplish tint near base. |
| 'Greray' | Sunray® | Medium | Symmetrical shape with an orange hue to young foliage. |
| Heavenly bamboo 'Lemon-Lime' | Lemon-Lime | Medium | Compact plant habit with chartreuse new foliage and contrasting green interior foliage. |
| Heavenly bamboo 'Murasaki' | Flirt™ | Dwarf | Compact, mounding habit with wine-red colored young foliage and grey green mature foliage. |
| Heavenly bamboo 'SEIKA' | Obsession™ | Medium-large | Densely foliated with bright red, young foliage that is retained while the plant is actively growing. |
| Heavenly bamboo 'Twilight' | Twilight | Dwarf-medium | Compact form with pink, young foliage and white variegation; mature foliage with green, pink, and white variegation. |
| Heavenly bamboo Wildtype | Heavenly bamboo | Large | Upright, rhizomatous shrub reaching 1.8–2.4 m tall with tripinnately compound, grey-green leaves turning reddish purple in winter. White terminal panicles beginning in May followed by globular red berries ripening in fall and persisting through the winter. |

[z] Overall plant sizes from field trials were used to assign height categories of dwarf (34.3–34.3 cm), medium (42.1–86.1 cm), or large (64.0–148.0 cm).

Maximum and minimum daily temperature at two meters, total rainfall, and relative humidity were recorded on site by the Florida Automated Weather Network (FAWN https://fawn.ifas.ufl.edu), as presented in S1 Fig. Prior to planting, soil samples were collected from each row at each site, mixed for uniformity, and air dried for standard analysis (UF Extension Soil Testing Laboratory, Gainesville, FL). Initial potassium (K), phosphorous (P), magnesium (Mg), and calcium (Ca) of soils based on Mehlich-3 extraction indicated sufficient nutrient ranges at all three field sites (S1 Table). The field conditions in southwest Florida were as follows: 2.14% organic matter, pH 6.35, electrical conductivity (EC) 0.05 dS/m, average monthly rainfall 11.43 cm, average monthly relative humidity 79.4%, average monthly temperature 25.8˚C, average monthly minimum temperature 21.5˚C, and average monthly maximum temperature 28.8˚C. The field conditions in northcentral Florida were as follows: 1.01% organic matter, pH 5.65, EC 0.10 dS/m, average monthly rainfall 9.7cm, average monthly relative humidity 81.1%, average monthly temperature 25.4˚C, average monthly minimum temperature 18.6˚C, and average monthly maximum temperature 33.1˚C. The field conditions in northern Florida were as follows: 2.1% organic matter, pH 5.35, EC 0.07 dS/m, average monthly rainfall 13.97 cm, average monthly relative humidity 80.6%, average monthly temperature 20.8˚C, average monthly minimum temperature 13.7˚C, and average maximum temperature 28.5˚C.

## Visual quality and plant growth

Assessments of foliage color and form (visual quality or plant performance) were performed in 3-month intervals at each site on a scale from 1 to 5 where 1 = very poor quality, not acceptable, severe leaf necrosis, 2 = poor quality, not acceptable or marketable, some areas of necrosis, poor form (irregular branching), 3 = adequate quality, somewhat desirable form and color, fairly marketable, 4 = good quality, very desirable color and form, and 5 = excellent quality, perfect condition, premium color and form. Plant size was measured every three months over a 79-week period by calculating growth indices as an average of the measured height (measured from crown to natural break in foliage) and two perpendicular widths [(width1 + width2)/2].

## Fruit production and seed viability

Every month the presence of flowering and fruiting of each plant was recorded at each site for the duration of the experiment. Before fruit ripening, mesh netting was placed over panicles to prevent predation. When most fruits were fully mature (second and third years of the study), they were manually harvested, and then counted at each location. Fruits were separated by color (mature vs immature) and then mature fruits were cleaned by hand using a dehulling trough (Hoffman Manufacturing, Inc., Albany, OR). Seeds were counted and those with insect or pathogen damage or abnormal appearance were noted. Seeds were stored in glass containers at room temperature until use.

Using fruit collected from the final (third) year of the landscape trials, seed viability tests were performed by an independent seed testing facility (US Forest Service National Seed Laboratory, Dry Branch, GA). A subsample of two replicates of 100 seed (when available) collected from the wildtype and 'Emerald Sea' plants (from each of the three locations) were subjected to a tetrazolium (TZ) staining test adapted from the Association of Official Seeds Analysts (AOSA) rules for Tetrazolium testing [14]. For very low fruiting cultivars ('Murasaki', 'SEIKA', and Twilight), all available, mature seeds were used for TZ testing without the ability to replicate. Seeds were cut laterally and stained overnight (12–18 h) at 37˚C in a 1.0% TZ solution. Seeds were considered viable when firm embryos stained evenly red.

## Nuclear DNA content and ploidy level

Young leaves were collected from heavenly bamboo plants (3 to 5 years old) grown at the GCREC (Balm). Three leaf samples (biological replicates) were analyzed per cultivar and replicated 3 times. Flow cytometry was performed as described by Doležel et al. [15] to determine nuclear DNA content and infer the ploidy. Methodology followed Wilson et al. [10] who reported inferred ploidy levels for other cultivars of heavenly bamboo. Young leaf tissue ($\approx$20 mg) was co-chopped with an equal amount of young reference tissue in 1 mL of the LB01 nuclei isolation buffer using a sharp razor blade, the released nuclei were stained with propidium iodide (50 µg/mL), the resultant nuclei suspension was filtered through a 50 µm pore size filter, propidium iodide was added, and the stained nuclei were analyzed on the flow cytometer Cyflow® Ploidy Analyzer (Sysmex Europe GmbH, Norderstedt, Germany) for fluorescence intensity. The LB01 buffer contained 15 mM Tris, 2 mM Na2EDTA, 0.5 mM spermine tetrahydrochloride, 80 mM KCl, 20 mM NaCl, and 0.1% (v/v) Triton X-100 and was adjusted to pH 7.5. Before use, RNase (New England BioLabs, Ipswich, MA) was added to the buffer to a final concentration of 50 µg/mL. Tomato (*Solanum lycopersicum* L. 'Stupické polní rané') (2C nuclear DNA content = 1.96 pg/2C) was used as an internal reference in the analysis [10,15].

## Experimental design and data analysis

The field experiments utilized a randomized complete block experimental design that was applied separately for each site. There were five blocks and eight treatments (seven cultivars and the wildtype heavenly bamboo), $n = 40$ at each of three locations (southwest, northcentral and north FL) for a total sample size, $N = 120$.

Data were analyzed using R (R.3.5.2, The R Foundation, Vienna, Austria) and RStudio (R 1.1.463, Boston, MA) linear mixed effects-models assuming normally distributed data. The assumptions for linear models were confirmed via QQ plots and plotting model residuals. No model selection was used since this was a planned experiment. Quality ratings measured across the entire experiment were modeled in response to cultivar, location, month, and all possible interactions with plot nested within block treated as a random effect. We analyzed quality data for year 1 and year 2 separately, and also for year 1 and year 2 combined. Height, width, and growth index at the end of the experiment (month 21) were modeled in response to cultivar, location, and cultivar*location interaction with experimental block being treated as a random effect. For all response variables, we then used Tukey's HSD to detect differences among treatment levels ($P \leq 0.05$) of statistically significant model terms.

## Results

### Landscape performance

Heavenly bamboo plant visual quality ratings varied by cultivar ($F_{7,92} = 19.78$; $P < 0.0001$) and location ($F_{2,92} = 19.03$; $P < 0.0001$) with a positive cultivar*location interaction ($F_{14,92} = 6.95$; $P < 0.0001$), revealing that cultivars responded differently across locations in the combined year 1 and tear 2 data (Table 2). Results were less clear for the data in year 1 alone. At southwest FL in year 1 mean visual quality was higher for 'SEIKA' (4.87) compared to 'Greray' (3.49), 'Lemon-Lime' (3.93), and 'Twilight' (3.73) but similar to that of 'Chime', 'Emerald Sea', 'Murasaki', and the wildtype (Table 3). In year 2, visual quality of 'Emerald Sea', 'Murasaki', and the wildtype were excellent (4.75 to 5.0) compared to 'Chime', 'Greray', 'Lemon-Lime', 'SEIKA' (that had adequate to good quality between 2.70 to 3.40) and 'Twilight' having poor

**Table 2. Linear mixed model results for analysis of the effects of location and cultivar on plant quality.**

| Year(s) | Treatment | DF num | DF den | F value | P |
|---|---|---|---|---|---|
| Year 1 | Cultivar | 7 | 92 | 1.66 | 0.1286 |
| | Location | 2 | 92 | 1.06 | 0.3515 |
| | Month | 3 | 283 | 32.72 | <0.0001 |
| | Location*Cultivar | 14 | 92 | 0.81 | 0.6530 |
| | Cultivar*Month | 21 | 283 | 3.25 | <0.0001 |
| | Location*Month | 6 | 283 | 0.51 | 0.7972 |
| | Location*Cultivar*Month | 42 | 283 | 1.28 | 0.1247 |
| Year 2 | Cultivar | 7 | 86 | 23.77 | <0.0001 |
| | Location | 2 | 86 | 55.40 | <0.0001 |
| | Month | 3 | 272 | 11.25 | <0.0001 |
| | Location*Cultivar | 14 | 86 | 7.40 | <0.0001 |
| | Cultivar*Month | 21 | 272 | 4.62 | <0.0001 |
| | Location*Month | 6 | 272 | 25.62 | <0.0001 |
| | Location*Cultivar*Month | 42 | 272 | 3.09 | <0.0001 |
| Year 1 & 2 | Cultivar | 7 | 92 | 19.78 | <0.0001 |
| | Location | 2 | 92 | 19.03 | <0.0001 |
| | Month | 6 | 550 | 16.16 | <0.0001 |
| | Location*Cultivar | 14 | 92 | 6.95 | <0.0001 |
| | Cultivar*Month | 42 | 550 | 4.16 | <0.0001 |
| | Location*Month | 12 | 550 | 35.40 | <0.0001 |
| | Location*Cultivar*Month | 84 | 550 | 3.26 | <0.0001 |

DF is degrees of freedom.

quality (1.70). Averaged over years 1 and 2, 'Emerald Sea', 'Murasaki', 'SEIKA' and the wild-type were the most attractive whereas 'Chime' and 'Twilight' were the least attractive (Table 3).

In northcentral FL during the first year, both the wildtype and cultivars had similarly very good to excellent visual quality ratings (ranging from 4.20 to 5.00 on a scale of 1 to 5) that were greater than 'Chime' that had poor to adequate quality ratings (2.63) (Table 3). This trend continued for the second year of the study where wildtype plants and all cultivars, except 'Chime', had high visual quality ratings (ranging from 4.50 to 5.00). Combined over both years plants had 1.7 times higher visual quality ratings (ranging from 4.49 to 4.94) than 'Chime' that had below average ratings (2.71).

In north FL, during the first year, both the wildtype and cultivars received adequate to good visual quality ratings (3.33 to 4.20 on a scale of 1 to 5) that were similar to each other but greater than 'Twilight' with average quality (3.20). This trend continued during the second year where the wildtype and all cultivars, except 'Chime' (4.30), had very good to excellent visual quality ratings (4.70 to 4.94) compared to 'Twilight' with lower quality ratings of adequate to good (3.80). Combined over both years, 'Murasaki' and 'SEIKA' had 1.2 and 1.3 times higher visual quality ratings as 'Chime' and 'Twilight', respectively.

## Plant size and growth

Seventy-nine weeks post planting final plant heights, widths and growth indices varied among cultivars and locations with significant main effects ($P<0.0001$) and their interaction for perpendicular widths ($F_{14,84} = 2.54$; $P<0.0045$), height ($F_{14,84} = 2.58$; $P<0.0039$) and growth index ($F_{14,84} = 1.81$; $P\leq0.0508$) (Table 4). Among cultivars, plant widths at each site ranged from

**Table 3. Mean plant quality ratings of eight heavenly bamboo (*Nandina domestica*) taxa during year 1, year 2 and across years 1 and 2.**

| | Plant quality rating (scale 1–5) | | |
|---|---|---|---|
| | Year 1[z] | Year 2[y] | Year 1 and 2[x] |
| Taxon | Mean ± SE | Mean ± SE | Mean ± SE |
| Southwest Florida | | | |
| Chime | 3.87 ± 0.35 abc | 2.70 ± 0.27 b | 3.20 ± 0.23 d |
| Emerald Sea | 4.60 ± 0.39 ab | 5.00 ± 0.18 a | 4.83 ± 0.22 a |
| Greray | 3.49 ± 0.22 bc | 3.40 ± 0.21 b | 3.44 ± 0.16 cd |
| Lemon-Lime | 3.93 ± 0.22 bc | 3.00 ± 0.05 b | 3.40 ± 0.11 cd |
| Murasaki | 4.40 ± 0.24 ab | 4.75 ± 0.11 a | 4.60 ± 0.12 ab |
| SEIKA | 4.87 ± 0.19 a | 3.40 ± 0.11 b | 4.03 ± 0.14 ab |
| Twilight | 3.73 ± 0.32 c | 1.70 ± 0.28 c | 2.57 ± 0.21 e |
| Wild type | 4.40 ± 0.13 ab | 4.75 ± 0.09 a | 4.60 ± 0.08 ab |
| Northcentral Florida | | | |
| Chime | 2.83 ± 0.39 bc | 2.63 ± 0.18 b | 2.71 ± 0.22 b |
| Emerald Sea | 4.20 ± 0.24 a | 4.70 ± 0.11 a | 4.49 ± 0.12 a |
| Greray | 4.35 ± 0.13 a | 4.80 ± 0.09 a | 4.61 ± 0.08 a |
| Lemon-Lime | 4.53 ± 0.18 a | 4.58 ± 0.11 a | 4.56 ± 0.1 a |
| Murasaki | 4.67 ± 0.27 a | 4.94 ± 0.06 a | 4.82 ± 0.13 a |
| SEIKA | 5.00 ± 0.16 a | 4.89 ± 0.07 a | 4.94 ± 0.08 a |
| Twilight | 4.67 ± 0.13 a | 4.50 ± 0.12 a | 4.57 ± 0.08 a |
| Wild type | 4.60 ± 0.22 a | 4.85 ± 0.08 a | 4.74 ± 0.11 a |
| North Florida | | | |
| Chime | 3.33 ± 0.22 ab | 43.0 ± 0.21 bc | 3.89 ± 0.16 b |
| Emerald Sea | 3.49 ± 0.19 a | 4.85 ± 0.11 ab | 3.96 ± 0.14 ab |
| Greray | 3.73 ± 0.17 a | 4.70 ± 0.11 ab | 4.29 ± 0.11 ab |
| Lemon-lime | 3.93 ± 0.15 a | 4.70 ± 0.13 ab | 4.37 ± 0.13 ab |
| Murasaki | 3.67 ± 0.32 a | 4.94 ± 0.06 a | 4.39 ± 0.17 a |
| SEIKA | 4.27 ± 0.17 a | 4.90 ± 0.00 a | 4.63 ± 0.10 a |
| Twilight | 3.20 ± 0.2 bc | 3.80 ± 0.21 c | 3.54 ± 0.17 cd |
| Wild type | 4.20 ± 0.19 a | 4.90 ± 0.07 a | 4.60 ± 0.11 a |

Qualitative scale (1 to 5) where 1 = very poor quality, 2 = poor quality, 3 = adequate quality, 4 = good quality, and 5 = excellent quality.

Different letters within columns are significantly different by Tukey–Kramer's honestly significant difference range test at $P \leq 0.05$.

[z]Year 1 data were collected from August 2019 to February 2020, beginning 3 months after planting.

[y]Year 2 data were collected from May 2020 to February 2021.

[x]Year 1 and 2 data are the means across both years.

25.73 to 80.70 cm (southwest FL), 31.90 to 94.17 cm (northcentral FL), and 42.65 to 88.32 cm (north FL) (Table 5). Plant heights ranged from 27.08 to 94.36 cm (southwest FL), 25.75 to 128.80 cm (northcentral FL), and 24.78 to 86.92 cm (north FL). Growth indices ranged from 26.46 to 87.53 cm (southwest FL), 28.83 to 111.49 cm (northcentral FL), and 35.34 to 85.66 cm (north FL). At all three sites, 'Emerald Sea' and the wildtype plants were among the widest and tallest compared to most cultivars with growth indices 1.1 times greater than medium-sized cultivars ('Greray', 'Lemon-Lime', 'Twilight', and 'SEIKA') and 1.63 times that of dwarf-sized cultivars ('Chime' and 'Murasaki') (Table 5). Also, among sites growth indices of 'Greray', 'Lemon-Lime', 'SEIKA', and 'Twilight' were nonsignificant from each other (except at the southwest location) and greater than 'Chime' and 'Murasaki'.

**Table 4. Results of linear mixed model analysis which examined the effects of location and cultivar on growth measurements (height, width, and growth index).**

| | Effect | DF num | DF den | F value | P |
|---|---|---|---|---|---|
| Width | Location | 2 | 84 | 71.87 | <0.0001 |
| | Cultivar | 7 | 84 | 53.33 | <0.0001 |
| | Location*Cultivar | 14 | 84 | 2.54 | <0.0045 |
| Height | Location | 2 | 84 | 25.21 | <0.0001 |
| | Cultivar | 7 | 84 | 79.84 | <0.0001 |
| | Location* Cultivar | 14 | 84 | 2.58 | 0.0039 |
| Growth index | Location | 2 | 84 | 44.59 | <0.0001 |
| | Cultivar | 7 | 84 | 94.29 | <0.0001 |
| | Location* Cultivar | 14 | 84 | 1.81 | 0.0508 |

DF is degrees of freedom.

## Flowering, fruiting, and DNA nuclear content

At all three locations, flowering was observed for 'Emerald Sea', 'Murasaki', 'SEIKA', 'Twilight', and the wildtype heavenly bamboo; typically began in April/May and lasted until June (data not presented). 'Emerald Sea' and the wildtype were the only two taxa that fruited at all three locations during both the second and third year of the study. In the second year across all taxa within locations, 'Emerald Sea' produced 25 (southwest), 126 (northcentral), and 132 (north) fruit when compared to wildtype plants that produced 293 (southwest), 363 (northcentral), and 917 (north) fruit. In the third year, this same cultivar produced 1549 (southwest), 1416 (northcentral), and 7756 (north) fruit when compared to wildtype plants that produced 1980 (southwest), 5159 (northcentral), and 5619 (north) fruit. This resulted in a 21.8%, 72.6% and 0% fruit reduction of 'Emerald Sea' relative to the wildtype for these sites (Table 6). For 'Murasaki', only one fruit was observed the second year from one location (northcentral FL). In the third year, three fruits were observed in southwest FL, two fruits were observed in northcentral FL, and 120 fruits were observed in north FL. This resulted in a 99.9%, 99.9%, and 97.7% fruit reduction relative to the wildtype in southwest, northcentral, and north FL, respectively, when compared to the wildtype plants at each location. For 'SEIKA', fruit was not observed in any location the second year. In the third year, fruit was also not produced in southwest FL, but

**Table 5. Average perpendicular plant width, plant height, and growth index of eight heavenly bamboo (*Nandina domestica*) taxa.**

| | Width (cm) | | | Height (cm) | | | Growth index (cm)$^z$ | | |
|---|---|---|---|---|---|---|---|---|---|
| Taxon | Southwest | Northcentral | North | Southwest | Northcentral | North | Southwest | Northcentral | North |
| Chime | 25.83 c | 31.90 d | 42.65 d | 27.08 c | 25.75 c | 28.02 c | 26.46 f | 28.83 d | 35.34 e |
| Emerald Sea | 80.70 a | 94.17 a | 88.32 a | 94.36 a | 128.80 a | 83.00 a | 87.53 a | 111.49 a | 85.66 a |
| Greray | 41.49 bc | 70.32 b | 60.03 cd | 56.74 b | 67.06 b | 54.02 b | 49.12 cd | 68.69 c | 57.03 cd |
| Lemon-Lime | 37.94 bc | 65.63 bc | 61.93 bc | 59.00 b | 70.70 b | 52.36 b | 48.47 cde | 68.17 c | 57.15 d |
| Murasaki | 35.60 bc | 37.81 cd | 46.83 cd | 28.06 c | 28.53 c | 24.78 c | 31.85 ef | 30.32 d | 35.80 e |
| SEIKA | 46.72 b | 76.87 ab | 69.73 bc | 82.80 a | 79.62 b | 65.70 ab | 64.76 bc | 78.25 bc | 67.72 bc |
| Twilight | 25.73 c | 69.40 bc | 52.31 cd | 56.68 b | 73.38 b | 56.76 b | 41.20 def | 71.39 c | 54.54 de |
| Wildtype | 65.99 a | 78.86 ab | 81.42 ab | 82.40 a | 109.7 a | 86.92 a | 74.20 ab | 94.28 ab | 84.10 a |

Plants grown in southwest Florida [Gulf Coast Research and Education Center (GCREC), Balm], northcentral Florida [Plant Science Research and Education Unit (PSREU), Citra], and north Florida [North Florida Research and Education Center (NFREC), Quincy] for 79 weeks.

Different letters within columns are significantly different by Tukey–Kramer's honestly significant difference range test at P≤0.05.

$^z$ Growth Index determined by (average of two perpendicular widths + height)/2.

**Table 6. Total fruit production (from five plants) in years 2 and 3 of heavenly bamboo (*Nandina domestica*) taxa grown in southwest FL (Balm), northcentral FL (Citra), and north FL (Quincy).**

| Taxon | Fruit no. year 2 | | | Fruit no. year 3 | | | Fruit reduction year 3 (%)[z] | | |
|---|---|---|---|---|---|---|---|---|---|
| | Southwest | Northcentral | North | Southwest | Northcentral | North | Southwest | Northcentral | North |
| Chime | 0 | 0 | 0 | 0 | 0 | 0 | 100.0 | 100.0 | 100.0 |
| Emerald Sea | 25 | 126 | 132 | 1549 | 1416 | 7756 | 21.7 | 72.6 | 0.0 |
| Greray | 0 | 0 | 0 | 0 | 0 | 0 | 100.0 | 100.0 | 100.0 |
| Lemon-Lime | 0 | 0 | 0 | 0 | 0 | 0 | 100.0 | 100.0 | 100.0 |
| Murasaki | 0 | 1 | 0 | 3 | 2 | 120 | 99.9 | 99.9 | 97.7 |
| SEIKA | 0 | 0 | 0 | 0 | 5 | 132 | 100.0 | 99.9 | 97.7 |
| Twilight | 0 | 0[y] | 0 | 0 | 0 | 230 | 100.0 | 100.0 | 95.9 |
| Wildtype | 293 | 363 | 917 | 1980 | 5159 | 5619 | NA | NA | NA |

Fruit typically have 1–2 seeds.

[z] Percent fruit reduction is calculated by [1-(no. cultivar's fruit/no. of wildtype's fruit)] * 100 based on respective wildtype fruiting at each site.

[y] Fruit observed in year 2 but did not reach maturity.

five fruits were observed in northcentral FL and 132 fruits were observed in north FL (Table 6). This resulted in a 99.9% and 97.7% fruit reduction relative to the wildtype in northcentral FL and north FL, respectively, when compared to the wildtype plants at each of these locations. For 'Twilight', fruits were not observed during the second or third year at southwest and central FL sites, but 230 fruits were observed in north FL (year 3). This resulted in a 95.9% fruit reduction of 'Twilight' when compared to the wildtype plants at the same location. 'Chime', 'Greray' and 'Lemon-Lime' did not fruit at any time or location during the study and were considered female sterile (100% fruit reduction) (Table 6).

Regardless of whether taxa fruited or not, nuclear DNA content ranged from 4.09 to 4.37 pg/2C among cultivars compared to the wildtype (4.07 pg/2C). This indicates that the heavenly bamboo taxa tested are diploids (Table 7).

Based on these assessments and use of a numerical fruiting scale (0–3), 'Chime', 'Greray', and 'Lemon-Lime' were categorized as 0 = nonfruiting, 'Murasaki', 'Twilight' and 'SEIKA' were categorized as 1 = low fruiting, and the wildtype and 'Emerald Sea' were categorized as 3 = heavy fruiting (Table 7). A moderate fruiting category was not observed among the cultivars in this study.

**Table 7. Fruiting categories, ploidy level, of eight heavenly bamboo (*Nandina domestica*) taxa.**

| Taxon | Fruiting category (0–3 scale) | Ploidy level | Nuclear DNA content ± SD (pg/2C) |
|---|---|---|---|
| Chime | 0 | $2x$ | 4.14 ± 0.15 |
| Emerald Sea | 3 | $2x$ | 4.19 ± 0.05 |
| Greray | 0 | $2x$ | 4.37 ± 0.02 |
| Lemon-Lime | 0 | $2x$ | 4.09 ± 0.06 |
| Murasaki | 1 | $2x$ | 4.32 ± 0.12 |
| SEIKA | 1 | $2x$ | 4.11 ± 0.11 |
| Twilight | 1 | $2x$ | 4.22 ± 0.07 |
| Wildtype | 3 | $2x$ | 4.07 ± 0.02 |

Scale where 0 = no fruiting, 1 = low fruiting, 2 = moderate fruiting, or 3 = heavy fruiting during the timeframe of the study.

Average nuclear DNA content ($n$ = 3) is presented ± standard deviation (SD).

**Table 8. Seed viability of five heavenly bamboo (*Nandina domestica*) taxa grown for 3 fall-winter seasons (124 weeks) in southwest FL, northcentral FL, and north FL.**

| | Southwest | Northcentral | North | Southwest | Northcentral | North |
|---|---|---|---|---|---|---|
| | No. seeds tested | | | Seed viability (%) | | |
| Emerald Sea | 145 | 200 | 15 | 8.3 | 29.0 | 6.7 |
| Murasaki | 5 | 5 | 3 | 0 | 40.0 | 33.3 |
| SEIKA | – | 6 | 56 | – | 66.7 | 33.9 |
| Twilight | – | – | 5 | – | – | 40.0 |
| Wildtype | 200 | 200 | 111 | 28.0 | 36.0 | 15.3 |

The no. of seeds tested value of 200 indicates two replicates of 100 seed were tested and fewer than 200 were treated as one replicate.

## Seed viability

Each fruit typically contained two seeds. The number of mature seeds available to conduct seed viability tests is shown in Table 8. Seed numbers were very low (less than six) for 'Murasaki' (from all three sites), 'SEIKA' (northcentral site), and 'Twilight' (north site), making it impossible to conduct replicated TZ tests for these cultivars. Nevertheless, seed viability of available seed varied widely among cultivars and locations, ranging from 0% to 66.7% compared to 15.3% to 36.0% for the wildtype (Table 8). Of the 'Emerald Sea' seeds, 8.3% were viable from the southwest location, 29.0% were viable from the northcentral location, and 6.7% viable from the north location. Of the 'Murasaki' seeds, 0%, 40.0%, and 33.3% were viable from southwest, northcentral, and north Florida locations, respectively. For 'SEIKA', there were no seeds available for TZ testing for the southwest location; and 66.7% and 33.9% seeds were viable from the northcentral and north locations, respectively. 'Twilight' only produced seeds in the north location that were 40.0% viable. Of the wildtype seeds, viability was 28.0%, 36.0%, and 15.3% from southwest, northcentral and north locations, respectively (Table 8).

## Discussion

Overall, at least half of the heavenly bamboo cultivars evaluated performed very well across Florida under full sun conditions and Florida's hot and humid summers. Cultivars that performed best (i.e., having quality ratings above 4.0) both years at all three locations included 'Murasaki' and 'SEIKA', having attractive reddish foliage on new growth. The wildtype similarly performed very well, regardless of year or location. This suggests the wide adaptability of these cultivars and the wildtype to temperature, soils, and harsh growing conditions with minimal inputs. In fact, the field site in northcentral FL had half the amount of organic matter than southwest or north Florida field sites, but this did not impact plant quality. 'Chime' and 'Twilight' typically underperformed in this study and may be more suitable to container or shaded conditions. The influence of geographic location on plant performance is consistent with prior studies that evaluated heavenly bamboo cultivars in south and north FL under similar full sun conditions [10,11]. In those studies, a limited proportion of cultivars (less than 25%) underperformed at one or both locations not meriting recommendation for landscape use, even if fruiting was absent.

Plant width, height and growth index measurements were useful in the overall categorization of plant size for heavenly bamboo. Plants were assigned to size categories, as large ('Emerald Sea' and wildtype), medium ('Greray', 'Lemon-Lime', 'SEIKA' and 'Twilight') or dwarf ('Chime' and 'Murasaki'). This information can be helpful when selecting plants for different areas of landscapes and gardens and may relate to plant vigor, a trait associated with invasiveness [16]. Geographical location (southwest, northcentral, or north FL) influenced the plant

size (growth index) of some cultivars. Plants tended to have moderately higher ('Emerald Sea') to slightly higher ('Greray', 'Lemon-Lime', 'SEIKA', 'Twilight', wildtype) growth indices at the northcentral location compared to the north or southwest locations. This effect of location on plant growth was not observed for dwarf forms ('Chime' and 'Murasaki'). Also of interest is that the wildtype and 'Emerald Sea' plants displayed cane-like growth qualities whereas this was absent in more densely foliated cultivars. Indeed, the wildtype can spread vegetatively from suckers and rhizomes, allowing it to form dense thickets that displace native vegetation [2,3]. This could be a relevant factor in an ITP assessment, as aggressive vegetative growth of sterile cultivars of another species, [Mexican petunia (*Ruellia simplex*)] led to its cautionary ITP conclusion (approved for use if managed to prevent escape) [17].

Flowering and fruiting were observed on 'Emerald Sea', 'Murasaki', 'SEIKA', 'Twilight', and the wildtype, but not every year and not at all locations. Flower abundance and duration are traits that have been associated with invasiveness. Gallagher et al. [18] reviewed data on 56 invasive species and 56 native species to Australia and found flowering of the invasive species to be one month longer than the native species, suggesting that longer flowering periods could allow for more pollinator visitation, seed set and increased propagule pressure. However, in the present study, and former heavenly bamboo studies [10,11] the onset and duration of flowering of wildtype plants was comparable to that of the cultivars.

In the timeframe of this study, fruit production of heavenly bamboo cultivars was absent ('Chime', 'Greray', and 'Lemon-Lime'), greatly reduced ('Murasaki', 'SEIKA' and 'Twilight'), or comparable ('Emerald Sea') to the wildtype taxon. Anecdotal information from Kluepfel and Polomski [1] claimed 'Murasaki', 'SEIKA', and 'Lemon-Lime' to be fruitless. We also did not observe fruit for 'Lemon-Lime', but after three years, a very small number of fruits were observed for 'Murasaki' and 'SEIKA'. There was a notable effect of location on fruiting during this study. In the cooler north FL location, fruiting of the wildtype taxon was more abundant (area annually receiving 420 chill hours) compared to northcentral (area annually receiving 110 but fewer than 420 chill hours) or southwest FL (area receiving 110 or fewer chill hours). This is consistent with a prior study that reported heavenly bamboo produced 9.8 times more fruit in north FL compared to south FL [10] and emphasizes the value of replicated trials in different geographic conditions as chilling hours are a requirement for some species. Additionally, the greater fruit production in north FL compared to northcentral and southwest FL emphasizes the importance of distinguishing the invasive status of plants independently for different regions of Florida, as exemplified by the *IFAS Assessment of Non-native Plants in Natural Areas*. To date, the heavenly bamboo wildtype taxon is only listed as invasive in central and north Florida [19].

Regardless of their capacity to fruit, both cultivated and wildtype forms of the heavenly bamboo that we evaluated were diploids (Table 8). Similarly, all 40 cultivars sampled from the JC Raulston Arboretum were found to be diploids [10,20]. Thus, polyploidy does not appear to be the driving factor behind the sterility of heavenly bamboo cultivars. In nature, polyploidy may confer advantages that could facilitate invasive potential such as faster growth and herbivore resistance [21]; and as such, polyploidy is one of the dataset variables used in invasive plant modeling [22]. Ploidy manipulation is a commonly used genetic approach to produce triploids that are often highly male- and female-sterile [23–25]. Using such an interploid hybridization system requires tetraploids that are either selected among existing cultivars or induced from diploids by chromosome doubling [26–28]. In addition to sometimes lengthy and expensive planned breeding programs to induce sterility in invasive ornamentals, naturally occurring whole plant mutations can be sources of novel and non-fruiting variants of heavenly bamboo.

Seed viability TZ testing is a destructive method to determine the potential for a seed lot to germinate, as it measures respiration [5]. In the present study, viability of the wildtype seeds ranged from 15.3% to 36.0% among locations. This is considerably lower than the viability of wildtype seeds reported in prior studies that was as high as 85.0% [11] and 86.5% [10]. The modest viability encountered in the present study may likely be a consequence of differences in plant maturity or environmental conditions such as temperature, soil chemistry, and rainfall in different growing regions over time. Qun et al. [29] describe three major deciding factors determining the level of seed vigor: the genetic constitution, the environment during development and the parameters of storage. More pertinent to this study, seed vigor may help with invasive plants' ability to outcompete and overtake an area.

Cultivars of heavenly bamboo were also found to have viable seed. Disagreement remains about what level of fecundity in cultivars can be tolerated without posing a risk to the environment. For instance, the Oregon Department of Agriculture approved seedless cultivars of a noxious weed, butterfly bush, for propagation, transportation, and sale provided that they produce less than 2% viable seeds [30]. Indeed, Knight et al. [31] raised the question of how much of a reduction in seed production or seed viability is necessary to create a cultivar that will not be invasive in natural areas. The authors emphasized that reduced seed production may be insufficient to eliminate the invasive potential of a species. Likewise, Bufford and Daehler [32] cautioned that horticultural selection for sterility (i.e., induced through transgenic techniques, through interspecific hybridization, or through chemically induced polyploidy to create triploid plants) can yield low-risk sterile cultivars of popular ornamentals provided that further hybridization or allopolyploidy does not restore fertility and vegetative spread is limited. Regardless, inducing sterility in plants is an ongoing effort of breeders and will continue to play a pivotal role in the ornamental industry. Certainly, intentionally bred sterile cultivars of ornamental plant species such as lantana (*Lantana strigocamara*) [33], Japanese barberry (*Barberis thunbergii*) [34], Norway maple (*Acer platanoides*) [35], and Mexican petunia (*Ruellia simplex*) [36] will ultimately decrease the propagule pressure and likelihood of invasion compared to the unregulated wildtype species. User-friendly resources and extension education will undoubtedly play a key role for consumers to distinguish between invasive and noninvasive cultivars as alternatives [37].

A notable limitation of the study was the inability to report the female fertility index of cultivars, where the number of fruits per peduncle is multiplied by the seed germination (in decimal form). The benefit of reporting the female reproductive potential of a given breeding line or cultivar is well described by Czarnecki and Deng [23]. Future studies should carefully consider counting the flowers and subsequent fruit produced per peduncle. Creating a female fertility index would help in answering the noteworthy questions pertaining to risk assessment posed by Datta et al. [26]. For example, what are the trait differences between cultivar alternatives and corresponding invasive species, how does this translate into differences in invasion risk and regulation and are these differences spatially and temporally stable. Another limitation of this study was the inability to conduct germination tests on a subsample of seeds to compare with viability tests either due to insufficient seed availability and/or the morphophysiological dormancy inherent with this species. Outside of this study, germination of heavenly bamboo seeds was found to be substantially delayed with the onset typically occurring around 77 d and extending to at least 168 d at 25/15˚C (unpublished data).

## Conclusions

In summary, we have evaluated plant performance and fruiting of the wildtype heavenly bamboo and seven cultivars at multiple locations in Florida. Based on these observations, 'Lemon-

Lime', 'SEIKA', 'Murasaki', and 'Greray' are good candidates for non-invasive status approval. Characteristics typical of these cultivars, such as desirable plant form, leaf morphology, color, and low to no fruiting, were consistent over time, with no observations of wildtype trait reversion. Thus, they are less likely to become invasive. These cultivars have been formally submitted to the UF/IFAS Assessment ITP and are pending approval. It is hoped that the promotion and wider use of these non-invasive cultivars can help reduce or eliminate the availability of heavily fruiting cultivars.

## Supporting information

**S1 Fig. Average monthly minimum and maximum temperatures, total rainfall (cm), and relative humidity (%), and soil recorded at trail site.** Sites located at southwest FL [Gulf Coast Research and Education Center (GCREC), Balm], northcentral FL [Plant Science Research and Education Unit (PSREU), Citra], and north FL [North Florida Research and Education Center (NFREC), Quincy]. Where week 0 starts in May 2019 and Week 82 ends in January 2020.
(TIF)

**S1 Table. Soil characteristics of trial sites located at southwest FL [Gulf Coast Research and Education Center (GCREC), Balm], northcentral FL [Plant Science Research and Education Unit (PSREU), Citra], and north FL [North Florida Research and Education Center (NFREC), Quincy].**
(TIF)

## Acknowledgments

We gratefully acknowledge Mark Kann, Keri Druffel, and Kelly Thomas, for their management of the research plots and assistance with data collection. We also thank James Colee for his assistance with the statistical analysis.

## Author Contributions

**Conceptualization:** Sandra Wilson, Zhanao Deng.

**Formal analysis:** Julia Rycyna, Basil Iannone.

**Investigation:** Julia Rycyna.

**Resources:** Gary Knox.

**Supervision:** Sandra Wilson, Zhanao Deng.

**Writing – original draft:** Julia Rycyna, Sandra Wilson.

**Writing – review & editing:** Julia Rycyna, Sandra Wilson, Zhanao Deng, Basil Iannone, Gary Knox.

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
