## [Decision Letter · Decision Letter 0]

18 Mar 2024

PONE-D-23-16117Landscape and female fertility evaluation of seven heavenly bamboo cultivars as potential non-invasive alternatives to the wildtypePLOS ONE

Dear Dr. Rycyna,

Thank you for submitting your manuscript to PLOS ONE. After careful consideration, we feel that it has merit but does not fully meet PLOS ONE’s publication criteria as it currently stands. Therefore, we invite you to submit a revised version of the manuscript that addresses the points raised during the review process.

We look forward to receiving your revised manuscript.

Kind regards,

Rajappa Janyanaik Joga, PhD

Academic Editor

PLOS ONE

Journal Requirements:

3. We note that your Data Availability Statement is currently as follows: "All relevant data are within the manuscript and its Supporting Information files."

Reviewers' comments:

Reviewer's Responses to Questions

**Comments to the Author**

1. Is the manuscript technically sound, and do the data support the conclusions?

Reviewer #1: Yes

Reviewer #2: Yes

2. Has the statistical analysis been performed appropriately and rigorously? 

Reviewer #1: Yes

Reviewer #2: Yes

3. Have the authors made all data underlying the findings in their manuscript fully available?

Reviewer #1: No

Reviewer #2: Yes

4. Is the manuscript presented in an intelligible fashion and written in standard English?

Reviewer #1: Yes

Reviewer #2: Yes

5. Review Comments to the Author

Reviewer #1: This study evaluates several new heavenly bamboo variety against wildtype over three years at multiple sites across Florida to evaluate their horticultural and invasive potential. The study achieves this well.

The abstract summarises main study results well but a little more context for the study would be useful. Is there a risk of invasiveness? Is the main aim to understand what causes reduced fecundity? The introduction gives good study context but some extra details could be included as suggested in specific comments. The study is incremental but the importance of monitoring new cultivars for their invasive potential is clearly described. The methods are mostly clear and comprehensive but a little more detail on the authors' definition of quality and form would be useful. The results are also comprehensive, but a few minor queries listed in specific comments remain. The discussion showed good interpretation and overview in terms of the aims of the study. There is useful consideration of both vegetative and sexual aspects of invasiveness. The value of multiple test sites across a climatic range are highlighted. The discussion on seed viability could also highlight the overall very low fruit number of most varieties. Ideas for future studies are suggested. The conclusions provide good overview to finish. Overall this is a well-presented study of relatively specialist interest but of regionally important value for plant invasion management.

Specific comments

L26-27 Perhaps add why reduced fecundity might sometimes be a desirable trait.

L43-45 One-third to two-thirds repeated as percentages.

L47-48 Mention DNA content measures and their reason earlier in the abstract methods.

L48-49 Name the four cultivars here.

L58 Add a comma between "India heavenly"

L62 It would be worth to add any observed negative ecological impacts of invasiveness.

L64 It is not clear to me from this description how morphology relates to dormancy.

L78 Explain to the reader why ploidy measurement is relevant to the study questions.

L97 It would be worth to briefly define what you mean by landscape performance here.

L114-115 Adding the size ranges for each classification would be even more helpful.

L121 What does "disked" mean in this context?

L122-124 What was the fumigant treatment targetting?

L136-147 Field conditions might be better presented in the results section.

L156-164 More specific detail on the scored parameters would be helpful. What is meant by "form" and what form characteristics merit different ratings.

L190-191 Repetition.

L223-225 Specify that this result applies to the combined year 1 and 2 data. It might be worth adding that results were less clear for year 1 alone.

L268-269 Clarify in the methods that growth index was a separate measure (mean of the two widths and height?)

L303 Specify that this fruit reduction is relative to wildtype.

L321 The formula needs to be multiplied by 100 to generate percentages.

From P19 line numbering lost.

P20 para 3. Specify if these seeds per fruit counts were from the x-ray analysis. If not, present some summary x-ray analysis results or omit this from the methods.

P21 para 1. The methods did not include germination tests.

P22 para 2. Did this study categorize the sizes of varieties or simply validate them based on previous knowledge?

The supplementary information misses the plant and seed measures.

Reviewer #2: In the realm of horticulture, this publication plays a crucial role by providing valuable insights for the industry. It serves as a window through which professionals can select appropriate specimens to meet their landscaping needs.

However, it is imperative that the authors adhere diligently to formal protocols when writing scientific names. For instance, consider the species "Nandina domestica Thunb.", which belongs to the family Berberidaceae. It is essential to recognize that this species hails from the berberis family and is not related to the grass family (Poaceae). Additionally, when referring to varieties, hybrids, cultivars, and mutants, strict adherence to the guidelines outlined in the International Code of Cultivated Plants Nomenclature (ICCPN) is essential.

In conclusion, the observations and experiments conducted suffice for now, but minor edits are necessary. Please refer to the uploaded PDF for specific revisions.

Congratulations on your work!

6. PLOS authors have the option to publish the peer review history of their article (what does this mean?). If published, this will include your full peer review and any attached files.

Reviewer #1: No

Reviewer #2: No

---

## [Author Response · Author response to Decision Letter 0]

1 May 2024

Reviewer 1 Comments

Line 26-27: Perhaps add why reduced fecundity might sometimes be a desirable trait.

This has been fixed.

Line 43-45: One-third to two-thirds repeated as percentages.

Fractions have been removed.

Line 47-48: Mention DNA content measures and their reason earlier in the abstract methods.

Thank you, we tried to rework the abstract by moving it earlier in the abstract; however, the flow of the abstract then did not match the flow of the methods and results. So we preferred to leave it were it is for consistency.

Line 48-49: Name the four cultivars here.

Cultivar names have been added.

Line 58: Add a comma between "India heavenly"

Comma has been added.

Line 62: It would be worth to add any observed negative ecological impacts of invasiveness.

Thank you, in lines 66 we added “displacing native species” to show negative ecological impacts

Line 64: It is not clear to me from this description how morphology relates to dormancy.

Thank you, Nandina has a rudimentary embryo and therefore classified as morphologically dormant. Morphological dormancy in this case is referring to the rudimentary embryo.

Line 78: Explain to the reader why ploidy measurement is relevant to the study questions.

We have included a sentence on line 84-86 to explain the importance of ploidy. In much of our work in breeding sterile cultivars for noninvasiveness, the ploidy level is triploid and not diploid.

Line 97: It would be worth to briefly define what you mean by landscape performance here.

Thank you, we clarified landscape performance on line 105 by adding the example of visual appeal. Landscape performance is already detailed with the visual quality scale listed on lines 165-172.

Line 114-115: Adding the size ranges for each classification would be even more helpful.

We have added size ranges in the footnotes.

Line 121: What does "disked" mean in this context?

By plowing the fields, roots of previous plant matter is dried out so that there is less plant competition.

Line 122-124: What was the fumigant treatment targeting?

The fumigant was multipurpose with the intention to sterilize the soil before planting. Soil was not tested for specific pathogens to target.

Line 136-147: Field conditions might be better presented in the results section.

Thank you, the environmental and soil condition at each site were characterized and objective of our study and therefore we included it as general methodology. 

Line 156-164: More specific detail on the scored parameters would be helpful. What is meant by "form" and what form characteristics merit different ratings.

Thank you we added that form is referring to irregular branching on line 174.

Line 190-191: Repetition.

The replicated sentence was removed and fixed on line 194. 

Line 223-225: Specify that this result applies to the combined year 1 and 2 data. It might be worth adding that results were less clear for year 1 alone.

This change has been made.

Line 268-269: Clarify in the methods that growth index was a separate measure (mean of the two widths and height?)

Growth index was calculated as two perpendicular widths [(width1 + width2)/2] we explained this in line 191-193 of the material and methods.

Line 303: Specify that this fruit reduction is relative to wildtype.

This change has been made.

Line 321: The formula needs to be multiplied by 100 to generate percentages.

The formula has been changed to generate percentages. 

From P19 line numbering lost.

Numbering was added to the remaining pages. 

Page 20 para 3: Specify if these seeds per fruit counts were from the x-ray analysis. If not, present some summary x-ray analysis results or omit this from the methods.

Good point the x-ray analysis was to determine a baseline on seed fill. The seed per fruit counts were not from the x-ray analysis. Results came from the tetrazolium tests so we removed the mention of x-rays from the methods section. 

Page 21 para 1: The methods did not include germination tests.

Germination was not mentioned in the methods because this study did not perform any germination. This study only looked at seed viability (destructive measure) because there were not enough seeds to perform both seed viability and germination. The words germination have been removed from the Seed Viability paragraph.

Page 22 para 2: Did this study categorize the sizes of varieties or simply validate them based on previous knowledge?

Size categories were determined based on previous knowledge from Wilson et al. 2014.

The supplementary information misses the plant and seed measures.

We only provided supplementary table for the soil conditions and a supplementary figure on the environmental conditions. Seed viability were provided directly in Table 8. The standard in the field is to report the processed data and not all the raw data needs to be presented.

Reviewer 2 Comments

In the realm of horticulture, this publication plays a crucial role by providing valuable insights for the industry. It serves as a window through which professionals can select appropriate specimens to meet their landscaping needs.

Thank you!

However, it is imperative that the authors adhere diligently to formal protocols when writing scientific names. For instance, consider the species "Nandina domestica Thunb.", which belongs to the family Berberidaceae. It is essential to recognize that this species hails from the berberis family and is not related to the grass family (Poaceae). Additionally, when referring to varieties, hybrids, cultivars, and mutants, strict adherence to the guidelines outlined in the International Code of Cultivated Plants Nomenclature (ICCPN) is essential.

In conclusion, the observations and experiments conducted suffice for now, but minor edits are necessary. Please refer to the uploaded PDF for specific revisions.

Line 27: complete the scientific name followed by its proper family designation 'Nandina domestica Thunb. (Berberidaceae)

This change has been made.

Line 29: heavenly bamboo cultivars

This change has been added.

Line 32: In northcentral Florida heavenly bamboo cultivars are ....

This change has been added.

Line 111: are this cultivated varieties, hybrids through natural selection or crossbreeds, need also to specify for clarity

We have changed the table title to include cultivars and the wildtype to explain that the plant studied are not hybrids or crossbreeds.

Line 112: This would be more clear if labeled as Cultivar Name

We have left the column as Taxon because it would be inaccurate to label it as cultivar name when there is no cultivar name for the wildtype. Instead we added a new column label common name to distinguish between cultivar names and trademark names that are commonly used. As per the ICBN general provisions for the name of cultivars we have added the unambiguous common name equivalent to accompany the cultivar epithet.

Line 475: Please elucidate the proper usage of what is considered a taxon and cultivars. Also refer to The International Code of Nomenclature for Cultivated Plants (ICNCP) how cultivar names are formally written. Also refer the legitimacy of the cultivar's registration used in the study.

As defined by the ICNCP, taxon is “a group into which a number of similar entities may be classified”. We decided it was appropriate to use this term because it would be incorrect to label them as cultivars since the wildtype is not a cultivar name. So, we decided to use the word cultivar when we only discuss cultivars and taxon when we are also discussing the wildtype. We refer to the plants by the cultivar epithet because the ICNCP allows it if it “is obvious from the context without confusion” and we have edited the description table to reflect the formal writing conventions. The cultivars chosen for this study were ones already available in the industry. Some cultivars are registered with US Plant Patents while others were not.

---

## [Editor Report · Decision Letter 1]

28 Aug 2024

Landscape and female fertility evaluation of seven heavenly bamboo cultivars as potential non-invasive alternatives to the wildtype

PONE-D-23-16117R1

Dear Dr.Julia Rycyna,

We’re pleased to inform you that your manuscript has been judged scientifically suitable for publication and will be formally accepted for publication once it meets all outstanding technical requirements.

Kind regards,

Rajappa Janyanaik Joga, PhD

Academic Editor

PLOS ONE
---

## [Editor Report · Acceptance letter]

13 Sep 2024

PONE-D-23-16117R1 

PLOS ONE

Dear Dr. Rycyna, 

I'm pleased to inform you that your manuscript has been deemed suitable for publication in PLOS ONE. Congratulations! Your manuscript is now being handed over to our production team.

Kind regards, 

on behalf of

Dr. Rajappa Janyanaik Joga 

Academic Editor

PLOS ONE